# Leveraging Class Hierarchy for Code Comprehension

**Jiyang Zhang, Sheena Panthaplackel, Pengyu Nie,**
**Junyi Jessy Li, Raymond J. Mooney, Milos Gligoric**

The University of Texas at Austin
{jiyang.zhang@, spantha@cs., pynie@, jessy@austin., mooney@cs., gligoric@}utexas.edu

## Abstract

Object-oriented programming languages enable a hierarchical class structure, which provides rich contextual information to guide code comprehension and synthesis. In this work, we propose the novel task of generating comments for overriding methods to facilitate code comprehension. To address this task, we formulate a deep learning framework which (1) exploits context from the comments of overridden methods and class names; (2) learns to generate comments in overriding methods that are more specific than those in the overridden methods; and (3) ensures that the generated comments are compatible with comments of overridden methods.

## 1   Introduction

The object-oriented paradigm enables developers to build hierarchical class structures. Under this paradigm, an *overriding* method m(args) is a method provided in a *subclass* $K^{\perp}$ with the same signature as a method m(args) (referred to as the *overridden* method) in class $K^{\top}$, which is the *superclass* of $K^{\perp}$. The overriding method $K^{\perp}$.m(args) is said to *override* the overridden method $K^{\top}$.m(args). For instance, in Figure 1, the InfoAccessSyntax class *inherits* members (methods and fields) from its superclass Object. Here, InfoAccessSyntax overrides Object's implementation of the getEncoded() method. It adds the functionality to initialize the encoding, i.e., when the encoding is null, the method will compute and use the ASN1 encoding.

The context provided by the class hierarchy can be useful for a wide variety of problems including code generation, code retrieval, and comment generation. For instance, the comment of the overridden method (Figure 1a) gives a general template which can be adapted to produce an accompanying comment for the overriding method (Figure 1b). As a step towards better understanding how to leverage class hierarchical context in deep learning models, we focus on generating comments for overriding methods in subclasses based on the comments of overridden methods.

Developers rely on natural language comments to understand important aspects of the source code they accompany, such as implementation, functionality, and usage. However, writing comments can be laborious and time-consuming, which is not ideal for fast-development cycles that are becoming increasingly prevalent (Panthaplackel et al. 2020; Liu et al. 2020; Hu et al. 2019; Tan et al. 2012). In practice,

```
public class Object {
  protected byte[] encoding;
  /** Returns encoded form of the object */
  public byte[] getEncoded() {
    return encoding;}}
```

(a) The getEncoded() method in the superclass, Object.

```
public class InfoAccessSyntax /* extends Object */ {
  /** Returns ASN.1 encoded form of this
      infoAccessSyntax */
  @Override public byte[] getEncoded() {
    if (encoding == null) {
      encoding = ASN1.encode(this);}
    return encoding;}}
```

(b) The getEncoded() method in the subclass, InfoAccessSyntax.

Figure 1: The InfoAccessSyntax class extends the superclass Object and overrides the getEncoded() method.

developers frequently just copy the comment from overridden method to overriding method, which results in a generic and non-descriptive comment for the overriding method. In this work, we aim to automatically generate comment suggestions that would help streamline the process of writing descriptive comments for overriding methods.

We design HIERARCHY-AWARE SEQ2SEQ, which leverages information from the class hierarchy. Namely, we incorporate a learned representation of the comment corresponding to the overridden method in the superclass, as this provides a general template that can be adapted. We also encode the class name, which often describes unique role and sometimes appears in the comment, as seen in Figure 1 (InfoAccessSyntax in the comment).

We further encode the notion of comment *specificity* (Ko, Durrett, and Li 2019; Zhang et al. 2018) to capture our observation that the comments for overriding methods usually contain specific words for describing their differences from the overridden methods, such as *"ASN.1"* in Figure 1. To ensure that the resulting lower-frequency words still pertain to the input, we also modify the architecture to encourage latent representations that are more similar to those of the input code and comments.

Additionally, we capture the intuition that the overriding

comments should be *compatible* to and should not contradict the comment corresponding to the overridden method. We introduce a *compatibility classifier* which allows us to give preference to generated comments that are compatible with their overridden counterparts during reranking.

Our main contributions are as follows: (1) we formulate the novel task of generating comments for overriding methods; (2) to address this, we design models that incorporate the class hierarchy in terms of context, specificity, and compatibility; (3) for training and evaluation, we build a large corpus of overriding-overridden method-comment tuples with their associated class hierarchy information.

## 2 Task

When a developer overrides a method, we aim to automatically generate a natural language comment which accurately reflects the method's behavior, capturing important aspects of the class hierarchy, as well as unique code that extends the implementation in the superclass. Concretely, in Figure 1, suppose a developer writes the method body of $M^\perp$ (getEncoded()) in class $K^\perp$ (InfoAccessSyntax) which overrides the parent method, $M^\top$, in the superclass, $K^\top$ (Object). Our task is to generate the comment for $M^\perp$, $C^\perp$ (*"Returns ASN.1 encoded form of this infoAccessSyntax"*), using context provided by the overriding method body, $M^\perp$, as well as the inputs from the class hierarchy. This includes the name of $K^\perp$, $Kname^\perp$; the name of $K^\top$, $Kname^\top$; the overridden method body, $M^\top$; and $C^\top$ (i.e., $M^\top$'s comment, *"Returns encoded form of the object"*).

We primarily consider generating the *first sentence* of the method-level comment, similar to prior work (Hu et al. 2019). The first sentence serves as a summary comment of the high-level functionality of the method. We also consider the more challenging task: generating the *full description* comment, i.e., the entire comment block without tags (e.g., @param). This includes both the high-level summary, as well as detailed descriptions of the method's functionality.

## 3 HIERARCHY-AWARE SEQ2SEQ

Our approach, HIERARCHY-AWARE SEQ2SEQ, leverages context from the class hierarchy to generate $C^\perp$. Shown in Figure 2, this is a SEQ2SEQ (Sutskever, Vinyals, and Le 2014) model that decodes $C^\perp$ using learned representations of three inputs: $M^\perp$, $Kname^\perp$, and $C^\top$ (§3.1). We also incorporate token-level auxiliary features into each of these encoders to further capture patterns pertaining to the class hierarchy, as well as properties of code and comments. During decoding, we discourage generating generic predictions by additionally injecting tailored representations that capture specificity (§3.2). Finally, we perform reranking to ensure that the prediction for $C^\perp$ is compatible with $C^\top$ (§3.3).

### 3.1 Encoders and Decoder

**Method Encoder**. We use a BiGRU (Cho et al. 2014) to encode $M^\perp$. Similar to prior work (Panthaplackel et al. 2020), we concatenate auxiliary features to the embedding vector corresponding to each input token before feeding it into the encoder. These include various features that capture whether

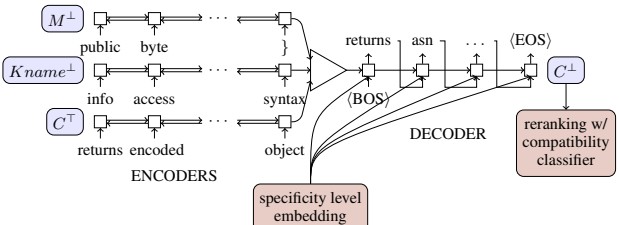

Figure 2: HIERARCHY-AWARE SEQ2SEQ architecture.

a token is a Java keyword or operator; appears only in $M^\top$, only in $M^\perp$, or both; and has lexical overlap in class names ($Kname^\perp$ and $Kname^\top$). These features are intended to help guide the model in correlating the method implementation with various components of the class hierarchy.

**Class Name Encoder**. Because $M^\perp$ is specific to subclass ($K^\perp$), the class name ($Kname^\perp$) can often shed light on $M^\perp$'s functionality. For instance, in Figure 1, knowing $Kname^\perp$ (InfoAccessSyntax) is helpful to generate the correct comment. Using a BiGRU, we learn a representation of $Kname^\perp$, as a sequence of subtokens. We similarly extract subtoken-level features which indicate whether a particular subtoken appears only in $Kname^\perp$ or also appears in $Kname^\top$. We also include features identifying lexical overlap between subtokens in $Kname^\perp$ and those in $M^\perp$.

**Superclass Comment Encoder**. We find the hierarchical relationship between $M^\perp$ and $M^\top$ to often hold between their accompanying comments. To provide context from the comment in the superclass, we introduce an encoder for learning a representation of $C^\top$. Like the other encoders, we concatenate features to the embedding vector corresponding to each token in the input sequence. We include features capturing lexical overlap with $Kname^\perp$ and $M^\top$ as we expect similar patterns to emerge in $C^\perp$. We also incorporate features that have been used to characterize comments in prior work (Panthaplackel et al. 2020): whether the token appears more than once, is a stop word, and its part-of-speech tag.

**Decoder**. We concatenate the final hidden states of each of the encoders to initialize the GRU decoder. We leverage attention (Luong, Pham, and Manning 2015) over the hidden states of all of these encoders. We additionally allow the decoder to copy tokens related to the implementation and class hierarchy from context provided by the inputs through a pointer network (Vinyals, Fortunato, and Jaitly 2015) over the hidden states of all three encoders.

### 3.2 Conditioning on Specificity

To encourage the decoder to predict a sequence that is specific and concretely reflects the functionality of $M^\perp$, we learn specificity embeddings (Ko, Durrett, and Li 2019). Namely, we discretize normalized inverse word frequency (Zhang et al. 2018) into 5 levels and associate an embedding with each. During training, each comment is assigned to its corresponding level, thus the embeddings are trained jointly with the model. At test time, following the intuition that overriding comments should be more specific, we feed in the embedding for the level that maximizes specificity at each time step.

| | Training | Validation | Testing |
|---|---|---|---|
| Projects | 414 | 16 | 41 |
| First sentence | 9,389 | 702 | 463 |
| Full description | 10,980 | 708 | 519 |

Table 1: Number of projects, first sentence, and full description examples used for the training/validation/test sets.

Because specificity alone would encourage the model to generate words of lower frequency, we try to encourage the model to prefer tokens that are semantically similar to the input. Specifically, we calculate coherence, which measures the similarity between the $C^\perp$ and $C^\top$ sentence embeddings. Sentence embeddings are computed as the weighted average of token embeddings, where the weights correspond to inverse document frequency. Similar to specificity, these coherence representations are also discretized into 5 levels, and their embeddings are jointly trained with the model. At test time, we select the maximum level for coherence.

### 3.3 Compatibility Reranking

We additionally ensure that the model prediction $C^\perp$ is *compatible* with $C^\top$; namely, $C^\perp$ should in principle be factually consistent with $C^\top$ and in many cases entail $C^\top$, e.g., Figure 1. To estimate such compatibility, we rely on the assumption that pairs of $(C^\top, C^\perp)$ are inherently compatible, and rerank candidate predictions using a pretrained compatibility classifier. This is a binary classifier trained to classify whether a subclass comment is compatible with a given superclass comment. We extract positive examples in the form $(C^\top, C^\perp)$ for every example in our training set. To form negative examples, we randomly select a subclass comment that does not belong to a subclass which inherits from $K^\top$, for every $C^\top$ in the training set. The classifier learns the representations of the two comments using GRU encoders and convolution layers. These two representations are concatenated and fed through a multi-layer perceptron network and a softmax layer to predict the compatibility label. The classifiers achieved 75.5% accuracy and 0.764 F1 score for first sentence comments; and achieved 71.5% accuracy and 0.742 F1 score for full description comments.

To rerank, we classify whether each beam search candidate is compatible with $C^\top$ and discard all incompatible ones, effectively moving up all compatible candidates. Our final model prediction is the highest ranked candidate. If all candidates are incompatible, we produce the candidate that was originally ranked highest.

## 4 Dataset

We build a corpus by mining open-source Java projects for examples in the form: $((M^\perp, C^\perp, K^\perp), (M^\top, C^\top, K^\top))$. From the Javadoc API documentation accompanying a given method, we derive comments from the main description, which precedes the tags (Oracle 2020b). As mentioned in Section 2, we consider both the *first sentence* of the main description part and the *full description*.

As done in prior work (Movshovitz-Attias and Cohen 2013; Panthaplackel et al. 2020), we partition the dataset in

such a way that there is no overlap between the projects used for training, validation, and testing. By limiting the number of closely-related examples across partitions, this setting allows us to better evaluate a model's ability to generalize. We filter out comments with non-English words, and remove those with $< 3$ words, as we find that these often fail to adequately describe functionality. We also discard trivial examples in which $C^\perp = C^\top$ as they lead to unwanted behavior in which the model learns to just copy $C^\top$. We tokenize source code and comments by splitting by space and punctuation and then split tokens of the form *camelCase* and *snake_case* to subtokens in order to reduce vocabulary size. Data statistics are given in Table 1. The average lengths of $C^\perp$ and $C^\top$ are 16.7 and 16.0 respectively for first sentence and 32.9 and 37.7 for full description comments.

## 5 Experiments

In this section, we describe several baselines, implementation details, and evaluation metrics.

### 5.1 Baselines

**COPY**. This is a rule-based approach which merely copies $C^\top$ as the prediction for $C^\perp$. This is in line with Javadoc tool which automatically copies $C^\top$ from $M^\top$ (Oracle 2020a).

**CLASS NAME SUBSTITUTION**. Based on our observations, there are many cases in which the developer obtains $C^\perp$ by simply copying $C^\top$ and replacing all occurrences of the parent class name with that of the child class. We simulate this procedure using rule-based string replacement: $C^\perp = C^\top.\texttt{Replace}(K^\top, K^\perp)$. Note that if the parent class name does not appear in $C^\top$, then $C^\perp = C^\top$.

**SEQ2SEQ**. We consider an approach that does not have access to class-related information. Namely, we use a SEQ2SEQ model with one BiGRU encoder for $M^\perp$, one GRU decoder for $C^\perp$, and attention (Luong, Pham, and Manning 2015) and copy (Vinyals, Fortunato, and Jaitly 2015) mechanisms.

**DEEPCOM-HYBRID (Hu et al. 2019)**. This approach uses two long short-term memory (LSTM) encoders to learn representations of the code and the flattened AST sequences. These are then used by an LSTM decoder to generate a sequence of comment tokens.

**EDIT MODEL**. We find that developers often produce $C^\perp$ by editing $C^\top$; however, these are not always as simple as class name substitution and require more complex edits. To address this, we include a model which *learns* to edit $C^\top$ in order to produce $C^\perp$. We adapt our recent comment editing framework that was originally proposed for updating comments that become outdated upon code changes to the corresponding methods (Panthaplackel et al. 2020). We first encode the existing comment using a BiGRU and the code edits with another BiGRU. We then use a GRU decoder to generate a sequence of comment edits which are applied to the existing comment. This leads to an updated comment that is consistent with the new version of the method. In our setting, we treat $C^\top$ as the "existing comment" and encode the code edits between $M^\top$ and $M^\perp$. We apply the generated comment edit sequence to $C^\top$ in order to produce $C^\perp$, which is expected to be consistent with $M^\perp$.

| Model | First sentence | | | Full description | | |
|---|---|---|---|---|---|---|
| | **BLEU-4** | **METEOR** | **ROUGE-L** | **BLEU-4** | **METEOR** | **ROUGE-L** |
| COPY | 24.228 | 19.845 | 43.352 | 20.623 | 18.757 | 40.846 |
| CLASS NAME SUBSTITUTION | 26.764 | 22.496 | 45.632 | 24.215 | 21.442 | 42.014 |
| SEQ2SEQ | 15.389 | 13.122 | 29.453 | 9.003 | 9.136 | 24.379 |
| DEEPCOM-HYBRID | 26.073 | 22.256 | 45.037 | 21.812 | 20.779 | 40.257 |
| EDIT MODEL | 31.755 | 26.926 | 46.939 | 23.734 | 22.571 | 44.359 |
| HIERARCHY-AWARE SEQ2SEQ | **35.899** | **31.131** | **53.548** | **27.719** | **26.908** | **49.352** |

Table 2: Comparison of our approach with baselines. Differences between all pairs are statistically significant.

## 5.2 Evaluation Metrics

Following prior work in comment generation and code summarization (Iyer et al. 2016; Hu et al. 2019; Liang and Zhu 2018; LeClair, Jiang, and McMillan 2019), we report metrics used to evaluate language generation tasks: BLEU-4 (Papineni et al. 2002), METEOR (Banerjee and Lavie 2005), and ROUGE-L (Lin 2004).

## 6 Results

We report results averaged across three random restarts. We use bootstrap tests (Berg-Kirkpatrick, Burkett, and Klein 2012) for significance testing under confidence level 95%. In Table 2, we present results for baselines and our HIERARCHY-AWARE SEQ2SEQ model. We first note that the COPY baseline underperforms the majority of other models (with the exception of SEQ2SEQ) across metrics for both datasets, demonstrating that generating the comment for an overriding method extends beyond simply repeating the overridden method's comment, contradicting prior claims (Hu et al. 2019). Both SEQ2SEQ and DEEPCOM-HYBRID, which do not have access to the context from the class hierarchy, underperform the other three approaches (besides COPY) that do have access to this information, including the rule-based CLASS NAME SUBSTITUTION baseline. This underlines the importance of context from the class hierarchy for generating overriding method comments. EDIT MODEL and HIERARCHY-AWARE SEQ2SEQ, the two models which *learn* to exploit the class hierarchical context, achieve higher performance than CLASS NAME SUBSTITUTION. We find HIERARCHY-AWARE SEQ2SEQ to yield better performance than EDIT MODEL. Recall that SEQ2SEQ's high-level neural composition closely matches that of HIERARCHY-AWARE SEQ2SEQ, only without any information pertaining to the class hierarchy. Hence, utilizing class hierarchy can achieve more than 80% improvement across metrics. While the results are analogous, performance across all metrics are consistently higher for *first sentences* than *full descriptions*, indicating, not surprisingly, that the latter is a more challenging task.

## 7 Related Work

Liu et al. (2019) address the task of comment generation and consider incorporating external context from the method call dependency graph by encoding the method names as a sequence of tokens. However, they do not consider the comments accompanying these methods. Haque et al. (2020) generate comments by encoding the method signatures of all other methods in the same file, again, without the accompanying comments. Rather than using the context of the call graph or other methods within the same file, we use the class hierarchy, and we also extract a comment (in the superclass) from this context, which we found to be one of the most critical components of our approach. Zhai et al. (2020) propose a rule-based approach for generating new comments by propagating comments from various code elements such as methods and classes. Such a system can generate a comment for the overriding method by simply propagating the comment of the overridden method (resembling our COPY baseline) or doing simple string replacements of class names (resembling our CLASS NAME SUBSTITUTION baseline). We instead *learn* how to leverage $C^\top$ in our HIERARCHY-AWARE SEQ2SEQ model, which outperforms both the COPY and CLASS NAME SUBSTITUTION baselines. Allamanis et al. (2015) study the task of method name generation. They build a log-bilinear neural language model which includes long-dependency contexts such as fields, sibling methods and super class to suggest method names. Their model learns the embeddings of names and output semantically similar names rather than learning to leverage hierarchy information to generate sequences as in our work.

## 8 Conclusion

In this work, we leverage the hierarchical class structure in object-oriented programming languages for automatic comment generation, aimed to guide code comprehension. We describe a new approach that utilizes this hierarchical structure, as well as specificity and compatibility, and show that it can be applied for automatically commenting overriding methods. Integrating this approach with the growing body of work in machine learning and natural language processing for software development will lead to the emergence of more intelligent, effective software engineering tools. Particularly, it would be interesting to adapt our hierarchy-aware approach to the inverse problem of program synthesis from natural language descriptions of overriding methods.

## Acknowledgments

We thank Darko Marinov, Thomas Wei, and the anonymous reviewers for their feedback on this work. This research was partially supported by the US National Science Foundation under Grant No. IIS-1850153, a Google Faculty Research Award, and a Bloomberg Data Science Fellowship.

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
