# OpenReview forum: "Leveraging Class Hierarchy for Code Comprehension"
_NeurIPS.cc/2020/Workshop/CAP — NeurIPS 2020 CAP Workshop_

### Official Review · AnonReviewer1 · 2020-10-30
**Very interesting idea**

**Rating:** 7
**Confidence:** 3

**Review:**

This paper concerns a specific language generation task, i.e. code comments prediction for object-oriented programs. The key observation is that class hierarchy usually provides rich contextual information, with which the authors developed a hierarchy-aware Seq2Seq model. The evaluation shows that hierarchy-aware Seq2Seq significantly outperforms the baseline, Seq2Seq, simple rule-based approach, as well as recent state-of-the-art models that do not consider class hierarchy information.

The writing is very clear. The presented idea is novel and experimental evaluations are convincing, which will encourage future work of leveraging domain-specific information to improve language generation in a specific context.

Questions:

Q1: How the dataset is collected? What are the factors considered for choosing an open-source Java project, e.g. popularity, size?

Q2: Have the authors considered the entire hierarchy tree, rather than the direct parent class?

---

### Decision · Program_Chairs · 2020-11-02

**Decision:**

Accept

**Comment:**

The review is quite positive, so I am recommending acceptance.